# OpenReview forum: "Scaling Long-Horizon Agent via Context Folding"
_ICLR.cc/2026/Conference — Submitted to ICLR 2026_

### Official Review · Reviewer_ZiqG · 2025-10-23

**Soundness:** 4
**Presentation:** 3
**Contribution:** 4
**Rating:** 6
**Confidence:** 4

**Summary:**

This paper introduces "Context Folding," a mechanism for LLM agents to manage their context by "branching" into sub-trajectories for specific subtasks and "returning" a summary, thereby "folding" the intermediate context. To train this behavior, the authors propose "FoldPO," an RL framework using process-level rewards (an "Unfolded Token Penalty" and an "Out-of-Scope Penalty") to guide the agent. The authors claim that on BrowseComp-Plus and SWE-Bench, their method (using a 32K active context) outperforms a 327K long-context ReAct baseline and a context summarization baseline.

**Strengths:**

1. The paper tackles a well-recognized and critical bottleneck in agent scalability. Managing context limitations is a key challenge for enabling more complex, long-horizon tasks.

2. The authors correctly identify that sparse task-level rewards (pass/fail) are insufficient to learn such a complex, hierarchical behavior. The design of the dense process rewards is clever and directly targets the two most likely failure modes: (1) failing to branch when needed (solved by the Unfolded Token Penalty) and (2) losing focus within a branch (solved by the Out-of-Scope Penalty). This process-oriented reward shaping is crucial for the method's success.

3. the empirical validation is comprehensive and convincing. The authors include the most critical baselines: (a) an agent with the same active context limit (ReAct 32K), (b) an agent with the same total token budget (ReAct 327K), and (c) a strong alternative approach (Context Summarization). The fact that the proposed method (with a 32K active context) outperforms the 327K long-context baseline is a significant result.

**Weaknesses:**

1. The "Out-of-Scope Penalty" relies entirely on a separate "GPT-5-nano" model to function. This may lead to reward hacking. Moreover, this may introduce bias originating from GPT-5-nano.

2. The context summarization baseline introduced in the paper is relatively simple, triggering only when the context is fully populated. I recommend introducing some other learning-based context summarization methods as baselines for comparison.

**Questions:**

see weakness

---

> ### Author Response · Authors · 2025-11-21
>
> Thank you for your thoughtful and valuable comment! It is encouraging to hear that you find our work addresses an important challenge, our method is clearly designed and crucial, and our experiments are comprehensive and convincing. We would like to address your concerns as follows:
>
> ---
>
> > **The "Out-of-Scope Penalty" relies entirely on a separate "GPT-5-nano" model to function. This may lead to reward hacking. Moreover, this may introduce bias originating from GPT-5-nano.**
>
> Thanks for the comment. First, we would like to clarify that the Out-of-Scope Penalty and Unfolded Penalty are supplementary mechanisms whose goal is to improve task performance. Our evaluation is based on a verified, unhackable task-success metric, and these penalties lead to substantial improvements in task success (Tables 1 and 2).
>
> To ensure reliable judging, we strengthened the process-reward judger. During training, we actively monitored GPT-5-nano’s explanations and added corrective examples to the judger prompt to address notable failure cases. Empirically, the judge behaves reasonably overall.
>
> Finally, we agree that some degree of reward hacking may still be unavoidable. We include these discussions in **Appendix G** and acknowledge that developing more reliable out-of-scope judgers remains an important direction for future work.
>
> ---
>
> > **The context summarization baseline introduced in the paper is relatively simple, triggering only when the context is fully populated. I recommend introducing some other learning-based context summarization methods as baselines for comparison.**
>
> Thank you for this suggestion. Context summarization is a simple yet highly effective method for agent context management. It is a widely adopted approach in many popular LLM agents, such as Claude Code, Manus, OpenHands, ReSum, MemAgent, and GPT-5-Codex.
>
> **In addition, we have carefully tuned this summary agent in preliminary experiments (details uploaded in Appendix F):**
> * **Prompts:** We ablated different summary prompts on BrowseComp to find the optimal one and reused the strong OpenHands condenser prompt for SWE-Bench.
> * **RL Setup:** We ablate different advantage estimators, and select the best-performing variant (sample-wise averaging with overlong masking).
>
> Therefore, we believe our baseline already represents the state-of-the-art in agent context management.
>
> We also note that some alternative memory methods are designed for **cross-session memory** (persisting knowledge across sessions) rather than **in-session context management** (managing context within a single session).
>
> ---
>
> **Thank you again for your thoughtful feedback! We hope these clarifications will resolve your concerns.**

---

> > ### Comment · Reviewer_ZiqG · 2025-11-26
> >
> > Thank you for the responses. While the clarifications are helpful, some of my main concerns remain partially addressed. Therefore, I will keep my original score.

---

> > > ### Author Response · Authors · 2025-11-26
> > >
> > > Thank you for your confirmation! We truly appreciate your feedback, which is invaluable for improving our paper.
> > >
> > > In addition, if you have any further questions or suggestions, we would be more than happy to discuss them. For example, if there are particular learning-based context summarization methods you would recommend for comparison, we would be glad to add a more detailed analysis.
> > >
> > > Best regards

---

### Official Review · Reviewer_SWnD · 2025-10-26

**Soundness:** 3
**Presentation:** 4
**Contribution:** 2
**Rating:** 4
**Confidence:** 3

**Summary:**

This paper tackles long-horizon LLM context management by actively controlling context rather than accumulating full histories. It introduces context folding: the agent uses tool tokens to branch into sub-trajectories with filtered, task-specific context, then returns a concise summary that is folded back into the main thread. To learn this behaviour, the authors propose FoldPO, an RL method that augments GRPO with token-level process rewards to encourage effective branching, scope adherence, and compact summarization. On deep-research and coding benchmarks, the approach matches systems with ~10× larger active context and outperforms models with similar context windows.

**Strengths:**

- The method is principled and applicable to a wide range of tasks.
- The method is general and principled  and appears applicable across tasks.

**Weaknesses:**

- Missing comparisons to closely related branching paradigms (e.g., Tree of Thoughts, Graph of Thoughts, “Everything of Thought”) to position the contribution.
- Some baselines are not directly comparable: fine-tuned models are compared against pre-trained long-context models. Where feasible, fine-tune the long-context baselines (e.g., GRPO) on the same benchmarks for a fair comparison.
- The method mainly **matches** the performance of models with much larger context windows rather than surpassing them. Since the introduction argues that long windows (i) can degrade performance and (ii) incur compute overhead, the paper should quantify compute/latency savings to justify the advantage of folding over long windows if (i) is not improved.
- Efficiency is under-reported. Please include metrics such as tokens read/generated and/or the average cost per query. It is unclear whether spawning additional branches increases total token usage and cost.

**Questions:**

- How does the plan-execution framework in 2.2 operate during RL training? What specific failure modes does it address?
- In FoldPO, how is the frequency of branch-token calls controlled or regularized? What prevents reward hacking via excessive branching, and conversely, collapse to zero branching if gains are marginal? Did you observe either behaviour in practice?

---

> ### Author Response · Authors · 2025-11-21
>
> Thank you for taking the time to review our paper and for your insightful comment! We’re glad to hear that you find our method principled and general. We would like to address your concerns as follows:
>
> ---
>
>
> > **Missing comparisons to closely related branching paradigms (e.g., Tree of Thoughts, Graph of Thoughts, “Everything of Thought”) to position the contribution.**
>
> Thank you for your thoughtful comment. We would like to clarify a key distinction. Tree-of-Thoughts–style methods (“X of Thoughts”) are **parallel sampling** paradigms, where multiple reasoning paths are generated to enhance reasoning. **In ToT, each reasoning path still operates with the full context.**
>
> Our method, in contrast, is a **context-management** technique for a single reasoning trajectory. It is not a parallel exploration strategy but a way to control context growth within one line of reasoning by folding parts of the history.
>
> **These approaches are thus orthogonal.** In tasks such as BrowseComp and SWE-Bench, the agent performs sequential multi-hop search or code read/edit/run steps, which can be viewed as a single path in a hypothetical ToT. In principle, one could incorporate context-folding into parallel sampling—for example, sampling multiple (intermediate) patches in SWE-Bench or multiple (intermediate) answers in BrowseComp and then selecting one with a value model, as in ToT.
>
> ---
>
> > **Some baselines are not directly comparable: fine-tuned models are compared against pre-trained long-context models. Where feasible, fine-tune the long-context baselines (e.g., GRPO) on the same benchmarks for a fair comparison.**
>
> We believe there are misinterpretations of our baseline setup. **Our main comparisons are against long-context baselines fine-tuned under strictly matched RL setup.** The long-context ReAct and Summary Agent baselines in Table 1 were both fine-tuned using GRPO on exactly the same data, token budget, and training setup as our method. All subsequent analyses compare against these RL-trained baselines and we always ensure apple-to-apple comparisons (RL vs RL, base model vs base model).
>
> We also report some 100B+ base models like GPT-5 in Table 1 as reference points that we cannot fine-tune, although they may already be optimized for search and coding tasks.
>
> ---
>
> > **The method mainly matches the performance of models with much larger context windows rather than surpassing them. Since the introduction argues that long windows (i) can degrade performance and (ii) incur compute overhead, the paper should quantify compute/latency savings to justify the advantage of folding over long windows if (i) is not improved.**
>
> > **Efficiency is under-reported. Please include metrics such as tokens read/generated and/or the average cost per query. It is unclear whether spawning additional branches increases total token usage and cost.**
>
> Our claim is supported by the evaluation results (Tables 1 and Figure 3, 5): **Our method substantially outperforms the RLed 320K ReAct baseline, especially on the most complex long-context tasks.** Specifically, our method scores 62% on BrowseComp-Plus compared to the baseline’s 54% **(+8%)**, and on the harder composed test set (answering multiple questions at once), the improvement is even larger (38% vs 69% **(+31%)**, Figure 5).
>
> Importantly, our method operates on **compressed context**, whereas ReAct uses the **full context**. These results indicate that current LLMs can experience performance degradation as context length increases.
>
> The performance gap is smaller on SWE-Bench because the tasks are less long-context-intensive, as most cases fit within 32K tokens.
>
> Regarding efficiency, we have added an analysis in **Appendix B**: our agent is **1.52× faster in end-to-end inference** and **1.43× faster in gradient updates** compared to the long-context ReAct baseline. Notably, this speedup is achieved despite our agent making 1.88× more tool calls on average (due to interaction scaling during RL training; see Figure 4), implying a per-tool-call speedup of roughly 2.86×.
>
> ---
>
> > **How does the plan-execution framework in 2.2 operate during RL training? What specific failure modes does it address?**
>
> The plan–execution framework has two key components:
> * First, the *prompt* instructs the model to reserve the main thread for high-level planning and to carry out concrete sub-tasks within branches.
> * Second, the *reward* applies an unfolded-token penalty to discourage excessive non-branch tool calls in the main thread.
>
> Beyond this *prompt* and *reward* shaping, the model is not subject to any additional architectural constraints. This design directly targets the failure mode in which the agent performs low-level work in the main thread, exhausts the main context, and undermines its ability to maintain a coherent high-level plan.
>
> ---
>
> (continue)

---

> > ### Author Response · Authors · 2025-11-21
> >
> > ---
> >
> > > **In FoldPO, how is the frequency of branch-token calls controlled or regularized? What prevents reward hacking via excessive branching, and conversely, collapse to zero branching if gains are marginal? Did you observe either behaviour in practice?**
> >
> > We do not apply any direct penalty or regularization on the frequency of branching. The unfolded-token penalty is applied only to excessive tool calls in the main thread. **The model cannot hack the reward through excessive branching**; that is, it cannot obtain more reward by creating more branches.
> >
> > **We also have not observed collapse to zero branching.** Empirically, the agent’s behavior is stable: the numbers of branches and tool calls gradually increase, possibly driven by the fact that more tool calls lead to higher task success, and more tool calls require more context and thus more branches.
> >
> > ---
> >
> > **Thank you again for your thoughtful feedback! We hope these clarifications will resolve your concerns.**

---

### Official Review · Reviewer_LcVw · 2025-11-01

**Soundness:** 3
**Presentation:** 3
**Contribution:** 2
**Rating:** 2
**Confidence:** 4

**Summary:**

The authors propose context folding, an agentic mechanism for managing long-horizon trajectories by selectively folding ephemeral sub-trajectories while preserving only essential decision-relevant information.
Additionally, they introduce FoldPO (a GRPO-style RL variant) with token-level process rewards (e.g., unfolded-token and out-of-scope penalties) to teach when to branch/return and what to retain.
On BrowseComp-Plus and SWE-Bench Verified, they report competitive pass@1 with a 32K active context (up to 10 branches), outperforming summarization baselines and approaching long-context (327K) ReAct agents.

**Strengths:**

* Ablations of the algorithms are informative.
* Clear problem framing (linear context growth), clean mechanism (branch/return), and simple plan–execute instantiation.
* Strong empirical performance at fixed active context; shows the benefit of learned context management vs. naive summarization.
* RL helps materially (FoldPO > vanilla GRPO); behavior analyses (more tool calls, more thorough exploration) are useful.
* Presentation is clear; diagrams/examples make the folding workflow easy to follow.

**Weaknesses:**

* Empirical validation lacks error bars; would encourage authors to always run experiments for multiple random seeds then report error bars.
* Missing direct comparisons to the most relevant recent systems: ReSum (periodic learned summarization), MemAgent/MEM1 (RL-trained memory/constant-size state). Without these, novelty/positioning is underspecified.
* Overclaims: “SOTA/comparable to 100B+” is too strong given GPT-5 baselines still lead; please tone down and specify precisely the scope (at fixed active context, same 36B base model, etc.).
* Summarization baseline may be underpowered vs. recent learned-memory methods; please ensure parity (tools, RL, prompts) or include those methods as baselines.
* No per-component reward ablation (e.g., remove unfolded-token penalty, remove out-of-scope penalty) to show each term’s necessity.
* No evaluation of summary fidelity (do fold summaries ever drop critical info?); even a small audit study or automatic factuality check would help.
* Minor inconsistency in reported numbers across text/table; ensure a single definitive pass@1 is reported and clarify variance across runs.
* Heavy reliance on an LLM judger (BrowseComp-Plus): discuss robustness/sensitivity; consider human spot-checking or multiple-judge aggregation.

**Questions:**

* Can you provide head-to-head results vs. ReSum and MemAgent/MEM1 under identical settings? If code/results are unavailable, a careful reimplementation of their summarization/memory policies would strengthen claims.
* Reward design ablation: what happens when each process reward is removed independently? Any degenerate behaviors (e.g., never returning, excessive branching)?
* Summary fidelity: do you have any quantitative/qualitative analysis showing critical facts survive folding? What failure modes did you observe?
* Please clarify the small discrepancy in BrowseComp-Plus pass@1 between text and table; also report mean±95% CI over multiple seeds.
* Compute/latency: how does rolling back KV-cache and creating branches affect wall-clock time vs. long-context and summarization baselines?
* Generality: outside web/coding, how does folding behave in dialogue or embodied tasks? Any constraints on nested tasks or multi-level folding?

---

> ### Author Response · Authors · 2025-11-21
>
> Thank you for your comment! We’re glad to hear that you find our experiments and analysis informative, our approach clear and competitive, and the paper easy to follow. We would like to respond as follows:
>
> ---
>
> > **Empirical validation lacks error bars; would encourage authors to always run experiments for multiple random seeds then report error bars.**
>
> Thanks for the suggestion! We have rerun our method under identical settings on BrowseComp-Plus twice, and the coverage test-set score lies between 0.62–0.64. We have updated the PDF to include the corresponding training details (See **Figure 9** in Appendix). Due to the high cost of model training, we are only able to run the baselines once; however, these runs are based on extensive preliminary experiments for debugging and hyperparameter tuning, which we believe leads to a stable baseline configuration.
>
> ---
>
> > **Missing direct comparisons to the most relevant recent systems: ReSum (periodic learned summarization), MemAgent/MEM1 (RL-trained memory/constant-size state). Without these, novelty/positioning is underspecified.**
>
> > **Summarization baseline may be underpowered vs. recent learned-memory methods; please ensure parity (tools, RL, prompts) or include those methods as baselines.**
>
> > **Can you provide head-to-head results vs. ReSum and MemAgent/MEM1 under identical settings? If code/results are unavailable, a careful reimplementation of their summarization/memory policies would strengthen claims.**
>
> Thanks for your comment. We believe there is a misunderstanding regarding our baseline design. ReSum/MemAgent/MEM1 focus on different tasks (BrowseComp, Ruler, HotpotQA) and use different backbones and agent scaffolds, so a direct model-to-model comparison is not feasible. We therefore reimplement their method on our dataset.
>
> We would like to clarify that we have already include a head-to-head comparison against ReSum/MemAgent/MEM1-based agent context management: our **“Summary Agent”** baseline (reported in Table 1 and Figure 3) is a **careful reimplementation** of their method (periodic, RL-trained summarization) in our dataset. We designed this baseline specifically to ensure a strict, apples-to-apples comparison: it uses the same base model, token budget, tools, training data, and RL infrastructure as our proposed method.
>
> **In addition, we have carefully tuned this summary agent in preliminary experiments (details uploaded in Appendix F):**
> * **Prompts:** We ablated different summary prompts on BrowseComp to find the optimal one and reused the strong OpenHands condenser prompt for SWE-Bench.
> * **RL Setup:** We ablate different advantage estimators, and select the best-performing variant (sample-wise averaging with overlong masking).
>
>
> To better clarify the implementation details, we have included **Table 3**, which provides a systematic comparison between ReSum/MemAgent/MEM1 and our method in terms of algorithmic design, model choices, and task settings.
>
> Finally, as we discuss (Line 241), **our conceptual novelty is distinct**: context folding is a learnable mechanism aligned with sub-task boundaries, preserving detailed reasoning until a sub-task is complete. This contrasts with periodic summarization, which discards details at arbitrary intervals.
>
> | | Ours | SummaryAgent | ReSum | MemAgent | MEM1 |
> |---|---|---|---|---|---|
> | Context | Folded Context | Summary | Summary | Summary | Summary |
> | Summary Trigger | Active | Passive | Passive | Every 5K-token| Every turn |
> | Active Context | 32K | 32K | 32K | 8K | ~1K |
> | Total Context | 320K (train) / 1M (test) | 320K (train) | Unknown | 32K (train) / 3.5M (test) | Unknown |
> | Model Size | 36B | 36B | 30B | 14B | 7B |
> | RL Algorithm | FoldPO | GRPO | GRPO (ReSum) | GRPO | PPO |
>
> ---
>
> > **Overclaims: “SOTA/comparable to 100B+” is too strong given GPT-5 baselines still lead; please tone down and specify precisely the scope (at fixed active context, same 36B base model, etc.).**
>
> We would like to clarify that we do not claim SOTA performance. For example, we have already avoided using words such as “SOTA”, “best”, or “top”. Our main performance claim is only that our method “*outperforms baselines that use 320K context*”, which is already scoped.
>
> Our statement that the method "*achieves performance comparable to agents built on much larger 100B+ parameter models*", is based on direct comparisons where our 36B agent outperforms several specific 100B+ models such as Qwen3-235B-A22B, GLM-4.5-Air, DeepSeek-V3.1, and GPT-4.1 on at least one of the two tasks (BrowseComp or SWE-Bench, see **Table 1**). These models are widely used as models for search and coding agents.
>
> We agree that GPT-5 is the strongest model on these benchmarks. To prevent any misunderstanding, **we have revised the paper to explicitly state:** "*Our 36B model is still behind the top-performing SOTA models like GPT-5*" and have ensured all claims are precisely scoped.
>
> ---
>
> (continue)

---

> > ### Author Response · Authors · 2025-11-21
> >
> > ---
> >
> > > **No per-component reward ablation to show each term’s necessity. Reward design ablation: what happens when each process reward is removed independently? Any degenerate behaviors (e.g., never returning, excessive branching)?**
> >
> > **Table 2 includes a reward ablation (removing both penalties)**, which demonstrates performance degradation. To further demonstrate the necessity of each component, **we have added the new ablation results to the appendix (see Figure 10)**, which ablate them independently.
> >
> > The two rewards are designed to be independent, as they apply to entirely different actions and tokens within the trajectory: the unfolded-token penalty applies to the main context to encourage the branch action, while the out-of-scope penalty applies to branch to encourage the branch action.
> >
> > **Our ablation results confirm this independence**: (1) Removing only the unfolded-token penalty causes a specific "context length exceed" degenerate behavior; (2) Removing only the out-of-scope penalty causes scope accuracy to collapse. **One penalty cannot fix the other's failure mode, proving both are essential.**
> >
> > ---
> >
> > > **No evaluation of summary fidelity (do fold summaries ever drop critical info?); even a small audit study or automatic factuality check would help. Summary fidelity: do you have any quantitative/qualitative analysis showing critical facts survive folding? What failure modes did you observe?**
> >
> > We have conducted a manual inspection of folded summaries. The summaries reliably and comprehensively capture the key search and exploration outcomes (e.g., crucial webpages or answers) and file edits. We have added qualitative examples in Table 5 illustrating how critical information is preserved across folds.
> >
> > Regarding failures or undesirable behavior, we did observe one: the agent sometimes includes redundant information from a previous branch in the current summary. While this redundancy is not encouraged, it does not materially harm downstream reasoning or final success.
> >
> > ---
> >
> > > **Minor inconsistency in reported numbers across text/table; ensure a single definitive pass@1 is reported and clarify variance across runs. Please clarify the small discrepancy in BrowseComp-Plus pass@1 between text and table; also report mean±95% CI over multiple seeds.**
> >
> > Thank you for pointing this out. We have fixed the inconsistency: the BrowseComp-Plus pass@1 score is **0.62**.
> >
> > ---
> >
> > > **Heavy reliance on an LLM judger (BrowseComp-Plus): discuss robustness/sensitivity; consider human spot-checking or multiple-judge aggregation.**
> >
> > We would like to clarify that the LLM judger we use is the official **reference-based judger**. It compares the model-predicted answer (an entity) and the ground-truth answer (an entity).
> >
> > To ensure robustness, during our experiments, we actively monitored the LLM-judge outputs and complemented them with a traditional Exact Match judger. **Through this process, we have identified and corrected three problematic ground-truth labels in the BrowseComp-Plus** (typos or entity aliases: "Tobias Smollet", "Biswaranjan Chattapadhyay", “Glafcos Clerides: The Path of a Country”); these corrections were verified with the BrowseComp-Plus authors. Outside of these three corrected errors, our manual audit found the LLM judger to be accurate.
> >
> > ---
> >
> > > **Compute/latency: how does rolling back KV-cache and creating branches affect wall-clock time vs. long-context and summarization baselines?**
> >
> > Rolling back does not introduce additional latency; it is simply reuse of a prefix in the KV cache and naive support by inference engine.
> >
> > Creating and returning branches does incur a modest overhead for generating the branch prompts and summaries, typically around **100 extra tokens per branch**. Figure 6 in the paper shows how the total token count increases during branching actions.
> >
> > We have also added **Appendix B** comparing actual inference and training time for our agent and the long-context ReAct baseline. Our agent is **1.52x faster** in end-to-end inference time compared to the long-context ReAct baseline. This speedup is achieved because our agent maintains a much shorter active context. Notably, since our agent makes 1.88x more tool calls on average (due to interaction scaling during RL training as shown in Figure 4), which implies a per-tool-call speedup of roughly 2.86×.
> >
> > ---
> >
> > > **Generality: outside web/coding, how does folding behave in dialogue or embodied tasks? Any constraints on nested tasks or multi-level folding?**
> >
> > We see no fundamental constraints. The context folding mechanism is general and does not require pre-defined nested tasks or multi-level hierarchies. It can be applied to any task (including dialogue or embodied AI) that parts of the trajectory can be compressed into shorter summaries without harming subsequent reasoning.
> >
> > ---
> >
> > **Thank you again for your thoughtful feedback! We hope these clarifications will resolve your concerns.**

---

### Official Review · Reviewer_HydY · 2025-11-01

**Soundness:** 3
**Presentation:** 3
**Contribution:** 3
**Rating:** 8
**Confidence:** 3

**Summary:**

The paper introduces Context Folding, a framework for long-horizon LLM operation that temporarily “branches” to perform token-intensive subtasks before folding the resulting information back into the main context via a compact summary, keeping the main thread manageable in scale. The core claim is that learning when to branch and how to summarize improves both quality and efficiency compared to ad hoc summarization prompting or simply scaling to longer contexts. This is paired with FoldPO, a variant of GRPO that uses token-level process rewards (a penalty for long main threads and a judge-based penalty for out-of-scope actions within branches). Empirically, on BrowseComp-Plus and SWE-Bench Verified, the method improves pass@1 on long-horizon benchmarks (e.g., browsing and code-fix tasks) while maintaining a much smaller active context than a long-context ReAct baseline and a context summarization method.

**Strengths:**

* **Originality.** Turning context management into an explicitly learned skill is a valuable idea at the intersection of hierarchical planning and summarization. Additionally, FoldPO's process-reward shaping directly targets the desired properties.
* **Quality.** Experiments cover multiple tasks and ablations, indicating that the different design choices matter.
* **Clarity.** The paper is well written and easy to follow; the figures effectively illustrate how branches are created/closed and how folded summaries condition subsequent actions.
* **Significance.** The approach can reduce reliance on very long contexts and enable more efficient agents, which constitutes a timely contribution.

**Weaknesses:**

* **Heuristic reward shaping.** The out-of-scope reward is provided by an auxiliary LLM judger. This risks bias/reward hacking. The unfolded-token penalty uses a fixed fraction of the working context budget, which seems very ad hoc and leaves room for designs that might lead to better learning.
* **Definition clarity.**
  * Equation (1) currently represents a probability over full traces at the action level, with the use of T that is dubious. Since the method in FoldPO is autoregressive at the token (or, if you prefer, action) level, I would encourage writing Eq. (1) in terms of the next-action distribution rather than the full-trace distribution, since I have doubts it is currently formally correct.
  * Q appears in the objective (L202) before it is defined. I encourage introducing Q immediately where first used.
* **Efficiency accounting.** Results emphasize *active* context compression. To highlight this, it would be relevant to include wall-clock latency and the total number of tokens processed (main thread \+ branches) per instance.
* **Formatting issues.** In Table 2, the value 78.10 should be (\<1) (likely 0.781).
* **Related work**. See the works below, which, although they do not subtract from the contributions of this work, which lies at the agency of context management, should be properly discussed in the related literature.

\[1\] Xu, B., Peng, Z., Lei, B., Mukherjee, S., Liu, Y., & Xu, D. (2023). ReWOO: Decoupling Reasoning from Observations for Efficient Augmented Language Models. *ArXiv*. [https://arxiv.org/abs/2305.18323](https://arxiv.org/abs/2305.18323)
\[2\] Holt, S., & Luyten, M. R. (2023). L2MAC: Large Language Model Automatic Computer for Extensive Code Generation. *ArXiv*. [https://arxiv.org/abs/2310.02003](https://arxiv.org/abs/2310.02003)
\[3\] Ning, X., Lin, Z., Zhou, Z., Wang, Z., Yang, H., & Wang, Y. (2023). Skeleton-of-Thought: Prompting LLMs for Efficient Parallel Generation. *ArXiv*. [https://arxiv.org/abs/2307.15337](https://arxiv.org/abs/2307.15337)
\[4\] Besta, M., Blach, N., Kubicek, A., Gerstenberger, R., Podstawski, M., Gianinazzi, L., Gajda, J., Lehmann, T., Niewiadomski, H., Nyczyk, P., & Hoefler, T. (2023). Graph of Thoughts: Solving Elaborate Problems with Large Language Models. *ArXiv*. [https://doi.org/10.1609/aaai.v38i16.29720](https://doi.org/10.1609/aaai.v38i16.29720)

**Questions:**

1. In eq 1, how is T formally defined? Would you be willing to refactor Eq. (1) into an autoregressive one?
2. You disabled nested branches for simplicity. Did you attempt it and observe failure modes when enabling shallow nesting? Although this is left as future work, preliminary results would, in my opinion, greatly increase the contribution.

---

> ### Author Response · Authors · 2025-11-21
>
> Thank you for your detailed and insightful comment! We are glad to know you find our method novel and timely, our experiment informative, and the overall presentation clear. We would like to address your concerns as follows:
>
> ---
>
> > **Heuristic reward shaping. The out-of-scope reward is provided by an auxiliary LLM judger. This risks bias/reward hacking. The unfolded-token penalty uses a fixed fraction of the working context budget, which seems very ad hoc and leaves room for designs that might lead to better learning.**
>
> Thank you for the comment. When implementing the out-of-scope reward, we monitored gpt-5-nano’s generated explanations during training and added several corrective examples to the judger prompt to handle notable failure cases. Empirically, the judge behaves reasonably overall. We agree that reward hacking or biased judging may still be unavoidable. We also agree that a more dynamic design for the unfolded-token penalty could improve the model’s flexibility. In future work, we plan to explore more robust approaches to address this issue. We have added the detailed judger design in **Appendix G**.
>
> ---
>
> > **Definition clarity.
> Equation (1) currently represents a probability over full traces at the action level, with the use of T that is dubious. Since the method in FoldPO is autoregressive at the token (or, if you prefer, action) level, I would encourage writing Eq. (1) in terms of the next-action distribution rather than the full-trace distribution, since I have doubts it is currently formally correct.
> Q appears in the objective (L202) before it is defined. I encourage introducing Q immediately where first used.**
>
> > **In eq 1, how is T formally defined? Would you be willing to refactor Eq. (1) into an autoregressive one?**
>
> Thanks for your suggestion! To clarify, in Equation 1 we use T to denote the number of turns (each agent-action and environment-observation pair, as shown in L107). Since the environment is deterministic, we express the full trace probability as the product of per-action probabilities. **To improve clarity, we have revised the paper to formally define T. We have also revised the paper around L202 to introduce Q immediately after its first use.**
>
> ---
>
> > **Efficiency accounting. Results emphasize active context compression. To highlight this, it would be relevant to include wall-clock latency and the total number of tokens processed (main thread + branches) per instance.**
>
> Thanks for your suggestion! We have added an efficiency analysis in Appendix B: our agent is **1.52× faster in end-to-end inference** and **1.43× faster in gradient updates** compared to the long-context ReAct baseline. Notably, this speedup is achieved despite our agent making 1.88× more tool calls on average (due to interaction scaling during RL training; see Figure 4), implying a per-tool-call speedup of roughly 2.86×.
>
> Regarding the total tokens processed for the main thread and branches, on average, out of **108.8K** total tokens, **7.8K** are processed in the main thread and **101.0K** are processed in the branches (including both model outputs and tool results).
>
> ---
>
> > **Formatting issues. In Table 2, the value 78.10 should be (<1) (likely 0.781).**
>
> Thanks for pointing this out. This is indeed a typo, we have corrected it.
>
> ---
>
> > **Related work. See the works below, which, although they do not subtract from the contributions of this work, which lies at the agency of context management, should be properly discussed in the related literature.**
>
> Thanks for your suggestion. We have revised the related work section to add discussion with these papers.
>
> ---
>
> > **You disabled nested branches for simplicity. Did you attempt it and observe failure modes when enabling shallow nesting? Although this is left as future work, preliminary results would, in my opinion, greatly increase the contribution.**
>
> Thank you for this insightful question. We did try nested branches on the base model, and our case studies showed that the model’s behavior was relatively hard to control; in particular, agents either did not create nested branches, or created them but never returned to the main thread. We also found that the task difficulty in BrowseComp or SWE-Bench does not seem to require such a sophisticated hierarchical structure.
>
> While we did not pursue nesting further, we tested a **parallel branching** approach in which the agent creates multiple branches that run simultaneously and aggregate results. We have added this discussion in **Appendix C**. The results show that this model achieves 0.6133 Pass@1 on BrowseComp-Plus, outperforming the long-context and summary agent baseline but performing similarly to the single-branch version.
>
> ---
>
> **Thank you again for your thoughtful feedback! We hope these clarifications will resolve your concerns.**

---

### Author Response · Authors · 2025-11-21

**We sincerely thank all reviewers for their insightful and constructive comments!**

We are encouraged to hear that reviewers recognize the core strengths of our work, including the clarity and principled nature of our approach, the value of learning context management as a capability, and the strong empirical evidence across tasks and ablations. We appreciate the positive remarks on our reward design, problem framing, and clear presentation, as well as the significance of enabling more scalable long-horizon agents.

We appreciate the invaluable suggestions that help us improve our paper. In response, we have addressed the following main concerns with corresponding revisions:

* **Comparison to the summary agent** (Reviewers LcVw, SWnD, ZiqG). We clarify the misunderstanding around our summary agent baseline, and include **Table 3** for a detailed comparison with related methods.

* **Missing efficiency analysis** (Reviewers SWnD, HydY). We add **Appendix B** with a model-efficiency study showing that our method is 1.43× faster in training compared to long-context ReAct.

* **Reward hacking concerns** (Reviewers LcVw, ZiqG, HydY). We clarify how our design mitigates reward hacking for the final reward, the out-of-scope judge, and the unfolded reward. These explanations are added to **Appendix G**.

We also revise the paper to fix typos and formatting issues, refine performance claims and related work, and add **Figures 9 and 10** for additional results, along with **Appendix C** for discussion on parallel branching. We have revised the paper accordingly, with all changes marked in red.

---

### Author Response · Authors · 2025-12-04

Dear AC,

Thank you for your time. Our paper unfortunately had no in-depth discussion before the OpenReview incident. We would like to briefly highlight that the key concerns, especially from the 2-score and 4-score reviewers, were based on misunderstandings that we have now resolved.
- Baseline misunderstanding clarified. Both reviewers believed we lacked comparisons to learned summarization/memory systems. We clarified that our summary agent baseline is already an RL-trained reimplementation of context-summarization methods, using identical model, token budget, tools, data, and RL setup. Table 3 and Appendix F document this clearly.
- Performance-claim misunderstanding clarified. We explained that our comparisons are strictly scoped (same 36B base, same RL setup) and that we notably outperform the long-context agent. We have also revised the paper to further prevent misunderstanding.

We also added:
- full efficiency metrics (1.52× faster inference, 1.43× faster training),
- independent reward ablations,
- variance across seeds,
- summary-fidelity inspection,
- judger robustness audit,
- clarified equations and fixed inconsistencies.

We hope this helps your assessment.

Sincerely,
The authors

---

### Meta-Review · Program_Chairs · 2026-01-05

**Summary:**

This paper proposes Context Folding, where an agent can branch into a sub-trajectory to handle a token-heavy subtask and then fold it back into the main thread via a concise summary, thereby keeping the active context small. The accompanying RL method, FoldPO, adds token-level process rewards to encourage branching when the main thread grows and to keep branch behavior on-scope. Reviewers agree the problem is important and the mechanism is clear; experiments on BrowseComp-Plus and SWE-Bench suggest competitive performance with much smaller active context, and the rebuttal adds efficiency numbers and additional ablations.

There exist two negative review. Personally I think the topic of this paper is quite important and new, and the paper provides clear motivation and training results. However, some notation and ablation are not clear, eg in figure there's mentioning of fold reward, which never appear in main paper; the paper didn't clearly show the role of process reward, and no comparison with standard reasoning RL with length-penalty to show advantage of such context folding, final performance comparison is also not conducted under same real-consumed token. Overall, this is a good direction but the paper need many careful revise and experiments.

**Reviewer Concerns:**

Technical details (efficiency, ablations, integration) were largely resolved. Remaining issues: clearly state of the difference against other refinement methods, and whether gains reflect deeper capability versus optimized sampling.

**Reviewer Scores:**

Score varied a lot for this paper; negative reviewer wants to add some standard summarizer baseline, while this paper is mainly for subagent rollout learning.

---

### Decision · Program_Chairs · 2026-01-26

Reject